# Instance-wise Knowledge Enhancement for 3D Instance Segmentation

## Abstract

Recent 3D Instance Segmentation methods typically follow a similar paradigm; they encode hundreds of instance-wise candidates with instance-specific information in various ways and refine them into final masks. However, they have yet to fully explore the benefit of these candidates. They overlook the valuable cues encoded in multiple candidates that represent different parts of the same instance, resulting in fragmented instance masks. Also, they often fail to capture the precise spatial range of complex 3D instances, primarily due to inherent fuzzy noises from sparse and unordered point clouds. In this work, to address these challenges, we propose **IKEA**, a novel instance-wise knowledge enhancement approach. We first introduce an *Instance-wise Knowledge Aggregation* to associate scattered single instance details by optimizing correlations among candidates representing the same instance. Moreover, we present *Instance-wise Structural Guidance* to enhance the spatial understanding of candidates using structural cues from ambiguity-reduced features. Here, we utilize a simple yet effective truncated singular value decomposition algorithm to minimize inherent noises of 3D features. Finally, our instance-wise features are now highly informative for real-world 3D instances. In our extensive experiments on large-scale benchmarks, ScanNetV2, ScanNet200, S3DIS, and STPLS3D, IKEA outperforms existing works. We also demonstrate the effectiveness of our modules based on both kernel and transformer architectures.

## 1 Introduction

3D Instance Segmentation (3DIS) is a fundamental 3D computer vision task that significantly contributes to real-world applications such as autonomous driving (Zhou et al., 2020) and robotics navigation (Xie et al., 2021a). Given 3D point cloud scenes, the 3DIS tasks identifies respective instances and assigns semantic class labels with the goal of comprehensively understanding entire spatial environments. In real-life 3D scenarios, substantial occlusion and truncation commonly arise, especially when objects are overlapped or obscured by others. To tackle these challenges, early approaches mainly concentrated on accurately generating region proposals (top-down) (Hou et al., 2019; Yang et al., 2019; Yi et al., 2019) or effectively grouping points with clustering algorithms (bottom-up) (Wang et al., 2018; Chen et al., 2021; Jiang et al., 2020). However, these methods presuppose that the intermediate processes, such as bounding box detection (He et al., 2017) or heuristic voting mechanism (Qi et al., 2019), generate near perfect results, which is often not the case.

Recently, kernel-based (Wu et al., 2022; He et al., 2021; 2022; Ngo et al., 2023) and transformer-based (Sun et al., 2023; Schult et al., 2022; Lai et al., 2023; Lu et al., 2023) 3DIS approaches have been proposed, aiming to overcome the limitations of traditional frameworks. Kernel-based methods leverage instance-aware kernels for dynamic convolution, which decodes instance masks. They represent instances as kernels, replacing clustering algorithms with point sampling processes. On the other hand, transformer-based methods train instance queries to identify features about individual objects using attention mechanisms. These queries enable the direct prediction of per-point categories and instance labels. In both architectures, models utilize hundreds of instance-wise candidate features, to estimate the final instance mask. For example, ISBNet (Ngo et al., 2023) (kernel-based) produces 256 instance kernels, and MAFT (Lai et al., 2023) (transformer-based) leverages 400 instance queries. However, despite the significant improvements based on their architectural advantages, they have yet to fully optimize the handling of these numerous candidates and still face several challenges, as illustrated in Fig. 1. To tackle these challenges, we introduce two main modules, IKA and ISG, each thoughtfully designed to enhance the informativeness of instance-wise candidates.

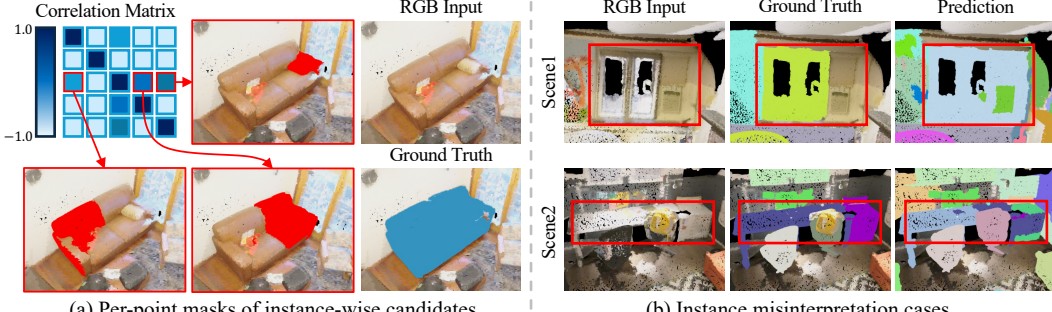

(a) Per-point masks of instance-wise candidates     (b) Instance misinterpretation cases

Figure 1: Examples of two challenging cases of 3DIS. (a) Per-point masks of instance-wise candidates: highly correlated instance-wise candidates usually represent incomplete fragments of the same single instance. And (b) Instance misinterpretation cases: existing studies often confuse instances from backgrounds (Scene1) or misunderstand the spatial range and appearance of instances (Scene2).

First, we empirically observe that multiple instance candidates from previous works often encode cues from the same single instance, as shown in Fig. 1 (a) and Fig. 9. These candidates are then typically decoded into considerable fragment masks representing different components for each instance, such as the backrest or armrest of the sofa. In this work, to estimate the complete coverage of each instance, we introduce a carefully designed *Instance-wise Knowledge Aggregation* (IKA) network, which integrates the scattered intelligence from multiple fragments, beyond relying solely on a few fragments based on confidence scores. Specifically, we optimize the correlations among instance-wise candidate features, encouraging the network to be aware of the distributed instance-specific knowledge. We initially identify candidates representing the same instance based on the similarity of their features, as highly correlated candidates usually capture the information from the same instance. Then, inspired by correlation regularization mechanisms (Zbontar et al., 2021; Bardes et al., 2021; Jang et al., 2024), we softly guide identified candidates to be closer to each other in a latent space, aiming to enhance solidarity among them. To this end, IKA enables the network to learn relationships between candidates and aggregate their comprehensive cues, maximizing the benefits of using hundreds of candidates. Through extensive experiments in Sec. 4 and Appendix C, we demonstrate the effectiveness of our proposed IKA network on 3D instance segmentation.

In addition to IKA, we explore additional avenues to enrich the knowledge of instance-wise candidates. Unlike the human visual cognitive system, which perceives instances with sufficient prior knowledge, the network understands instances depending on their external properties from 3D features. However, point features are frequently unstable, primarily due to ambiguous inherent noises (Feng et al., 2021; Xie et al., 2021b; Wu et al., 2021; Ren et al., 2021) caused by sparse and incomplete point clouds. Consequently, these noises are likely to confuse the models, making it difficult to precisely interpret the specific spatial range and unique appearance of instances, as shown in Fig. 1 (b) and Fig. 10 . To mitigate the negative impact of such noises, we propose an *Instance-wise Structural Guidance* (ISG) network, which is specifically designed to improve the structural understanding of instance candidates for 3D instances from point features. Here, we utilize the truncated singular value decomposition (SVD) (Golub & Reinsch, 1971; Stewart, 1993) algorithm to strengthen essential cues (*e.g.*, shape) of instance features while explicitly reducing fuzzy noises. Then, we effectively transfer fundamental clues for object cognition from ambiguity-minimized features to corresponding original features based on the cross-correlation matrix, as in the IKA. Ultimately, our instance features are now highly informative across complex 3D instances with aggregated and structural knowledge from IKA and ISG. Note that our novel networks can be applied to methods that use considerable instance-wise candidates, regardless of model structures, including kernel and transformer-based architectures.

Given landmark datasets for 3DIS, ScanNetV2 (Dai et al., 2017), ScanNet200 (Rozenberszki et al., 2022), S3DIS (Armeni et al., 2016), and STPLS3D (Chen et al., 2022), we thoroughly validate the effectiveness of our novel framework, IKEA. Above all, our method outperforms the existing state-of-the-art methods. To summarize, our main contributions are listed as follows:

- To extend the benefits of producing multiple instance-wise candidates, we introduce the Instance-wise Knowledge Aggregation (IKA) network, which associates scattered instance-specific information of the same single instance by optimizing correlations among them.

- We design the Instance-wise Structural Guidance (ISG) network to improve the structural knowledge of candidates. Specifically, we use a simple yet strong truncated singular value decomposition process to emphasize essential cues from features while filtering out noises.

- We analyze the effectiveness of our proposed methods on four challenging datasets, including ScanNetV2, ScanNet200, S3DIS, and STPLS3D. Comprehensive experiments across various 3D scenarios demonstrate that ours achieves new State-of-the-Art performance on 3DIS.

## 2 RELATED WORK

**3D Instance Segmentation (3DIS)**   aims to distinguish individual instances within 3D scenes and assign corresponding semantic categories. Typically, 3DIS approaches can be categorized into four types: proposal-based (Hou et al., 2019; Yang et al., 2019; Yi et al., 2019), grouping-based (Wang et al., 2018; Elich et al., 2019; Chen et al., 2021; Engelmann et al., 2020; Jiang et al., 2020), kernel-based (Wu et al., 2022; He et al., 2021; 2022; Ngo et al., 2023), and transformer-based (Sun et al., 2023; Schult et al., 2022; Lai et al., 2023; Lu et al., 2023). Based on the impressive achievements of detection methods (He et al., 2017; Wang et al., 2022), proposal-based approaches first detect instances and employ proposed regions as hard references to predict masks. However, these strategies, such as 3D-SIS (Hou et al., 2019) or 3D-BoNet (Yang et al., 2019), heavily rely on the quality of the outputs from the detection process. On the other hand, grouping-based methods catch 3D instances through a bottom-up pipeline, aggregating closely related points into instances using predicted semantic categories and center offsets. Yet, they still depend on intermediate manually tuned processing, such as point grouping (Jiang et al., 2020) or heuristic voting mechanisms (Qi et al., 2019), to specify detailed geometric properties. To address these limitations, kernel-based approaches have been introduced. For example, DyCo3D (He et al., 2021) employs a clustering algorithm to generate kernels for dynamic convolutions to predict instance masks, while ISBNet (Ngo et al., 2023) presents a cluster-free method using instance-wise kernels. Recently, Spherical Mask (Shin et al., 2023) based on ISBNet has overcome the low-quality results of coarse-to-fine strategies by using spherical representation to mitigate the propagation of false negatives and instance size overestimation issues. Furthermore, transformer-based methods, Mask3D (Schult et al., 2022) and MAFT (Lai et al., 2023), utilize instance-wise queries to encode information about individual objects based on attention mechanisms. In this work, we focus on improving the quality of numerous instance-wise candidates from recent studies and introduce novel instance-wise knowledge enhancement approaches.

**Singular Value Decomposition (SVD)**   is a fundamental algorithm in linear algebra, widely used for matrix factorization (Golub & Reinsch, 1971; Stewart, 1993; Klema & Laub, 1980), dimensionality reduction (Wall et al., 2003; Yang et al., 2014; Howland et al., 2003), and 2D image processing (Rajwade et al., 2012; Guo et al., 2015; Dian et al., 2020; Shi et al., 2021). SVD decomposes a matrix $A \in \mathbb{R}^{m \times n}$ into the dot product of three matrices, $U \in \mathbb{R}^{m \times m}$, $\Sigma \in \mathbb{R}^{m \times n}$, and $V \in \mathbb{R}^{n \times n}$, each containing meaningful vectors and scalar values, as follows:

$$A = U \cdot \Sigma \cdot V^T \tag{1}$$

where $U$ and $V$ are the matrices with orthonormal columns, satisfying $U^T U = V^T V = I$, and $\Sigma$ is a diagonal matrix with nonnegative singular values $\sigma$ in descending order as $\sigma_1 \geqslant \sigma_2 \geqslant \cdots \geqslant \sigma_r$. Based on SVD, truncated SVD (Hansen, 1987) is an insightful dimension-reduction technique that minimizes nonessential noise while highlighting essential cues from the original matrix $A$. It produces a subset of the most important $k$ singular values to derive a low-rank approximation $\tilde{A} \in \mathbb{R}^{m \times n}$ as:

$$\tilde{A} = U_k \cdot \Sigma_k \cdot V_k^T \tag{2}$$

where $U_k \in \mathbb{R}^{m \times k}$, $\Sigma_k \in \mathbb{R}^{k \times k}$, and $V_k \in \mathbb{R}^{n \times k}$ are the truncated matrices with only top $k$ values.

In the field of 2D image processing, SVD has been extensively utilized for low-rank approximation. For instance, Rajwade et al. (2012); Guo et al. (2015) employ SVD to factorize and estimate low-rank patches, effectively reducing noise and enhancing image quality. Similarly, Chang et al. (2005); Kang & Wei (2008) operate SVD to detect tampering by extracting and analyzing low-dimensional representations of images for image forensics. Also, SVD showcases robust performance in image compression (Prasantha et al., 2007; Bryt & Elad, 2008) and recovery (Shi et al., 2021). Further, Dian et al. (2020) leverages SVD within CNNs to learn subspace information for hyperspectral image

Figure 2: An overview of IKEA framework. Built upon the modern kernel-based structure, IKEA consists of four main modules: (1) Sparse Convolutional 3D Backbone; (2) Instance-wise Knowledge Aggregation (IKA, Sec. 3.2), which softly associates scattered cues from highly correlated instance features of $F_{inst}$; (3) Instance-wise Structural Guidance (ISG, Sec. 3.3), which smartly instructs $F_{inst}$ with essential clues from $\tilde{F}_{inst}$, using an effective SVD algorithm; and (4) Dynamic Convolution.

reconstruction. However, despite these potential benefits, the introduction of SVD has received limited attention in 3D vision, especially for point clouds. Based on these observations, we utilize SVD to address point features containing inherent noises from unordered properties of point cloud data, facilitating the network to become more robust against various ambiguities. To the best of our knowledge, we are the first to introduce the usefulness of SVD in the feature space of 3DIS.

# 3    METHOD

In this section, we introduce a novel 3D Instance Segmentation (3DIS) framework, IKEA, which (1) integrates scattered instance-specific knowledge across multiple instance-wise candidates and (2) enhances the structural understanding of candidates with essential cues from noise-reduced features. We first provide an overview of the whole pipeline (Sec. 3.1) and then present technical details: (Sec. 3.2) Instance-wise Knowledge Aggregation and (Sec. 3.3) Instance-wise Structural Guidance.

## 3.1    OVERVIEW

Recent 3DIS frameworks (He et al., 2021; 2022; Ngo et al., 2023; Schult et al., 2022; Lai et al., 2023; Lu et al., 2023) commonly follow a similar paradigm; they spread hundreds of instance-wise candidates across 3D scenes to capture instance-specific information in various ways and refine them into a final instance mask. In this work, we encourage these numerous instance candidates to be highly informative, improving their understanding of diverse real-world instances. We describe our instance-wise knowledge enhancement approaches based on the modern kernel-based architecture (Ngo et al., 2023). As illustrated in Fig. 2, our model consists of four main modules: (1) Sparse Convolutional 3D Backbone; (2) Instance-wise Knowledge Aggregation network, which relates the scattered information of individual instances; (3) Instance-wise Structural Guidance network, which effectively instructs candidates with spatial cues from augmented instance features; and (4) Dynamic Convolution network. Note that we also describe **transformer-based** IKEA architecture in the Appendix B.

**Kernel-based 3DIS Architecture.**    First, sparse convolutional U-Net backbone (Graham et al., 2018) takes a colored point cloud $P \in \mathbb{R}^{N_p \times 6}$ as input and voxelizes $P$ into voxels to extract point-wise feature maps $F_p \in \mathbb{R}^{N_p \times D}$. Given the point feature, the point aggregator samples a set of instance candidate features, referring to point positions based on the Farthest Point Sampling (FPS) (Eldar et al., 1997). Specifically, we employ two-stage point aggregator blocks to produce optimal instance features across 3D scenes. In the first stage, $N_k'$ number of candidates $F_{inst}' \in \mathbb{R}^{N_k' \times D}$ are sampled from full-resolution feature $F_p$, while in the second stage, $N_k$ ($< N_k'$) instance features $F_{inst} \in \mathbb{R}^{N_k \times D}$ are sampled based on the $F_{inst}'$. We then classify the category $C_i$ for $i = \{1, 2, \ldots, N_c\}$ of each instance from $F_{inst}$ via linear classification head $f_{cls}$ using following cross-entropy loss:

$$\mathcal{L}_{cls} = -\mathbb{E}_{c, w_c \sim \mathbb{D}} \left[ \sum_{r \in N_c} w_c[r] \log f_{cls}(F_{inst})[r] \right] \qquad (3)$$

Figure 3: Instance-wise Knowledge Aggregation: Based on the correlation matrix $I_{ka}$, we specify highly correlated instance candidates ($I_{ka} > \tau$) representing identical instances. Then, we softly aggregate scattered clues of related candidates, optimizing $I_{ka}$ with pseudo-binary label $I_{ka}^*$.

Figure 4: Instance-wise Structural Guidance: We decompose $F_{inst}'$ into three matrices ($U$, $\Sigma$, $V$) using the SVD algorithm and keep top $t$ singular vectors to filter out less important noises. Then, we effectively enhance $F_{inst}$ with highlighted essential cues from $\tilde{F}_{inst}$ by optimizing $\mathcal{L}_{SG}$.

where $w_c$ indicates one-hot encoded category labels and $\mathbb{D}$ represents (input) data distribution. Also, we predict instance kernels $K \in \mathbb{R}^{N_k \times D}$ based on $F_{inst}$ through MLPs. Furthermore, each point-wise prediction head processes $F_p$ to generate mask features $F_{mask} \in \mathbb{R}^{N_p \times D}$ and axis-aligned bounding boxes $F_{box} \in \mathbb{R}^{N_p \times 6}$ as auxiliary information for mask prediction. Finally, the dynamic convolution network produces the final instance mask $\hat{M}_f$ using instance kernels $K$ and $F_{mask}$. To optimize the final mask, we formulate the following loss $\mathcal{L}_{mask}$ as a sum of the two losses $\mathcal{L}_{BCE}$ and $\mathcal{L}_{dice}$:

$$\mathcal{L}_{mask} = \lambda_{BCE}\mathcal{L}_{BCE}(M_f, \hat{M}_f) + \lambda_{dice}\mathcal{L}_{dice}(M_f, \hat{M}_f) \tag{4}$$

where $M_f$ denotes the ground-truth instance mask, and $\mathcal{L}_{dice}$ represents Dice loss (Deng et al., 2018).

## 3.2 INSTANCE-WISE KNOWLEDGE AGGREGATION (IKA)

Recent studies (Wu et al., 2022; He et al., 2021; 2022; Ngo et al., 2023; Schult et al., 2022; Lai et al., 2023; Lu et al., 2023) in 3DIS deal with instance-wise features $F_{inst}$ as essential components, aiming to encode rich semantic and geometric information about instances. They spread hundreds of candidates over expansive 3D scenes to cover as many instances as possible; this number $N_k$ far exceeds the total number $N_i$ of instances present in any single scene (see Appendix Tab. 12). Although these candidates provide practical information for mask prediction, they often represent separate parts of each instance, resulting in fragmented instance masks. With these insights, we focus on associating scattered knowledge to fully leverage the advantages of multiple instance candidates.

To handle multiple candidates, we first identify those representing the same instance based on their pairwise affinity. Specifically, we compute a correlation matrix $I_{ka} \in \mathbb{R}^{N_k \times N_k}$ using the dot product of $F_{inst}$ and its transpose $F_{inst}^T$, then normalize $I_{ka}$ to a range between 0 and 1. Here, we note that candidates representing the same instance yield relatively high correlation values. Thus, based on $I_{ka}$, we consider pairwise candidates with correlation values surpassing a predefined threshold $\tau$ as fragments from the same instance. Then, as shown in Fig. 3, we softly guide these candidates by optimizing their correlations in $I_{ka}$, motivated by the success of recent self-supervised learning strategies (Zbontar et al., 2021; Bardes et al., 2021). We extend this approach to further enrich coherence among highly correlated candidate features while also encouraging the network to be aware of scattered cues for each instance. In particular, we dynamically construct a pseudo-binary correlation label matrix $I_{ka}^* \in \{0,1\}^{N_k \times N_k}$, where elements of highly correlated pairs ($> \tau$) in $I_{ka}$ are 1, while the rest ($\leq \tau$) are 0 for all candidate pairs. Given $I_{ka}$ and $I_{ka}^*$, we formulate the element-wise correlation regularization loss $\mathcal{L}_{KA}$ as follows:

$$I_{ka} = F_{inst} \cdot F_{inst}^T, \quad I_{ka}^*(i,j) = \begin{cases} 1 & \text{if } I_{ka}(i,j) > \tau \\ 0 & \text{otherwise,} \end{cases} \tag{5}$$

$$\mathcal{L}_{KA} = \sum_{I_{ka} > \tau} \|I_{ka}^*(i,j) - I_{ka}(i,j)\|_2 + w \cdot \sum_{I_{ka} \leq \tau} \|I_{ka}^*(i,j) - I_{ka}(i,j)\|_2 \tag{6}$$

where $\|\cdot\|_2$ denotes the $L_2$ norm, and $w$ is the weight to balance between highly correlated pairs and others. In Eq. 6, the first term facilitates candidates representing the identical instance closer together

in latent space, whereas the second term decreases correlations with unrelated candidates, thereby reducing confusion from irrelevant knowledge. In conclusion, IKA establishes valuable connections that strengthen solidarity across multiple candidates with scattered cues by regularizing $\mathcal{L}_{KA}$.

### 3.3 INSTANCE-WISE STRUCTURAL GUIDANCE (ISG)

In real-world scenarios, 3D instances are usually placed near each other or overlapping, arranged in inconsistent patterns without conventional standards. Also, these instances are represented as sparse point clouds, which often induce fuzzy inherent noises in point-wise features. Therefore, the instance-wise features $F_{inst}$ from these settings are unclear, resulting in misinterpretation of the spatial extent of each instance. In this work, we explicitly reduce such ambiguity of $F_{inst}$ by leveraging intelligent structural guidance from clarity-enhanced instance-wise feature $\tilde{F}_{inst}$.

For accurate segmentation, it is crucial to capture the clear morphological shape of complex instances. From this intuition, we aim to encourage the model to learn the structural appearance of instances better. Thus, we implement effective structural guidance based on a truncated singular value decomposition (SVD) (Stewart, 1993; Golub & Reinsch, 1971) algorithm primarily used for noise reduction. As illustrated in Fig. 4, given $F'_{inst} \in \mathbb{R}^{N'_k \times D}$ from the first point aggregator, we initially decompose $F'_{inst}$ into three matrices based on SVD: (1) $U \in \mathbb{R}^{N'_k \times N'_k}$, (2) $\Sigma \in \mathbb{R}^{N'_k \times D}$, and (3) $V \in \mathbb{R}^{D \times D}$, as:

$$F'_{inst} = U \cdot \Sigma \cdot V^T = \sum_{i}^{N'_k} u_i \cdot \sigma_i \cdot v_i^T \tag{7}$$

where $U = (u_1, \ldots, u_n)$ and $V = (v_1, \ldots, v_n)$ are the matrices with orthonormal columns, and $\Sigma = diag(\sigma_1, \ldots, \sigma_n)$ is a diagonal matrix containing singular values, arranged in the descending order. Typically, the higher singular values in $\Sigma$ include meaningful information for object cognition, while the lower-rank vectors are regarded as less important, potentially causing ambiguity. Then, we preserve only the top $t$ singular values, along with their corresponding singular vectors from $U$ and $V$, facilitating an ambiguity-reduced yet information-rich representation of the original feature $F'_{inst}$ as:

$$\tilde{F}'_{inst} \in \mathbb{R}^{N'_k \times D} = \tilde{U} \cdot \tilde{\Sigma} \cdot \tilde{V}^T = \sum_{i}^{N'_k} \tilde{u}_i \cdot \tilde{\sigma}_i \cdot \tilde{v}_i^T \tag{8}$$

where $\tilde{U} \in \mathbb{R}^{N'_k \times t}$, $\tilde{\Sigma} \in \mathbb{R}^{t \times t}$, and $\tilde{V} \in \mathbb{R}^{D \times t}$ are truncated matrices using only $t$ columns from $U, \Sigma$, and $V$, then $\tilde{F}'_{inst}$ is reconstructed from them. Based on both $F'_{inst}$ and $\tilde{F}'_{inst}$, we sample instance-wise features $F_{inst}$ and $\tilde{F}_{inst}$, respectively, employing the shared second stage point aggregator. Given $F_{inst}$ and $\tilde{F}_{inst}$, we calculate the cross-correlation matrix $I_{sg} \in \mathbb{R}^{N_k \times N_k}$ and also generate a pseudo-binary correlation label $I^*_{sg} \in \{0, 1\}^{N_k \times N_k}$, as in the IKA network. We then effectively transfer structural clues from $\tilde{F}_{inst}$, utilizing the correlation regularization loss $\mathcal{L}_{SG}$ as follows:

$$I_{sg} = F_{inst} \cdot \tilde{F}_{inst}^T, \quad I^*_{sg}(i,j) = \begin{cases} 1 & \text{if } I_{sg}(i,j) > \tau \\ 0 & \text{otherwise}, \end{cases} \tag{9}$$

$$\mathcal{L}_{SG} = \sum_{I_{sg} > \tau} \left\| I^*_{sg}(i,j) - I_{sg}(i,j) \right\|_2 + \tilde{w} \cdot \sum_{I_{sg} \leqslant \tau} \left\| I^*_{sg}(i,j) - I_{sg}(i,j) \right\|_2 \tag{10}$$

where $\tilde{w}$ represents the balancing hyper-parameter, and the role of each term in Eq. 10 follows Eq. 6. By optimizing $\mathcal{L}_{SG}$, ISG conditionally instructs instance-wise features $F_{inst}$ to incorporate spatial configuration details from clarity-enhanced features $\tilde{F}_{inst}$, thereby improving discriminative power.

**Loss Function.** Finally, our framework is trained by minimizing the following loss function $\mathcal{L}_{total}$:

$$\mathcal{L}_{total} = \lambda_{mask}\mathcal{L}_{mask} + \lambda_{cls}\mathcal{L}_{cls} + \lambda_{KA}\mathcal{L}_{KA} + \lambda_{SG}\mathcal{L}_{SG} \tag{11}$$

where each $\lambda$ is a hyper-parameter from grid searches to handle the strength of respective loss term.

## 4 EXPERIMENTS

### 4.1 EXPERIMENTAL SETUP

**Datasets.** In this study, we train and evaluate the overall performance using four landmark benchmarks for 3D instance segmentation: ScanNetV2 (Dai et al., 2017), ScanNet200 (Rozenberszki et al.,

Table 1: Comparison of 3DIS performance with state-of-the-art approaches on the ScanNetV2 (Dai et al., 2017) validation set.

| | ScanNetV2 | | |
|---|---|---|---|
| Method | mAP | $mAP_{50}$ | $mAP_{25}$ |
| GSPN | 19.3 | 37.8 | 53.4 |
| 3D-SIS | – | 18.7 | 35.7 |
| MTML | 20.3 | 40.2 | 55.4 |
| 3D-MPA | 35.5 | 59.1 | 72.4 |
| DyCo3D | 35.4 | 57.6 | 72.9 |
| PointGroup | 34.8 | 56.7 | 71.3 |
| MaskGroup | 42.0 | 63.3 | 63.3 |
| OccuSeg | 44.2 | 60.7 | 71.9 |
| SSTNet | 49.4 | 64.3 | 74.0 |
| SoftGroup | 46.0 | 67.6 | 78.9 |
| Mask3D | 55.2 | 73.7 | 82.9 |
| QueryFormer | 56.5 | 74.2 | 83.3 |
| ISBNet | 56.8 | 73.3 | 81.3 |
| MAFT | 58.4 | 75.9 | 84.5 |
| Spherical Mask | 62.3 | 79.9 | 88.2 |
| IKEA (Ours) | **62.9** | **81.8** | **88.7** |

Table 2: Comparison of 3DIS performance with state-of-the-art approaches on the S3DIS (Armeni et al., 2016) Area 5 and 6-fold cross validation. For clarity and readability of Tab. 1 and Tab. 2, we describe each method with references in Appendix C.1.

| | S3DIS Area 5 | | | | S3DIS 6-fold CV | | | |
|---|---|---|---|---|---|---|---|---|
| Method | AP | $AP_{50}$ | $Prec_{50}$ | $Rec_{50}$ | AP | $AP_{50}$ | $Prec_{50}$ | $Rec_{50}$ |
| SGPN | – | – | 36.0 | 28.7 | – | – | 38.2 | 31.2 |
| ASIS | – | – | 55.3 | 42.4 | – | – | 63.6 | 47.5 |
| 3D-Bonet | – | – | 57.5 | 40.2 | – | – | 65.6 | 47.6 |
| 3D-MPA | – | – | 63.1 | 58.0 | – | – | 66.7 | 64.1 |
| PointGroup | – | 57.8 | 61.9 | 62.1 | – | 64.0 | 69.6 | 69.2 |
| DyCo3D | – | – | 64.3 | 64.2 | – | – | – | – |
| MaskGroup | – | 65.0 | 62.9 | 64.7 | – | 69.9 | 66.6 | 69.6 |
| SSTNet | 42.7 | 59.3 | 65.5 | 64.2 | 54.1 | 67.8 | 73.5 | 73.4 |
| SoftGroup | 51.6 | 66.1 | **73.6** | 66.6 | 54.4 | 68.9 | 75.3 | 69.8 |
| ISBNet | 56.3 | 67.5 | 70.5 | 72.0 | 60.8 | 70.5 | 77.5 | 77.1 |
| DKNet | – | – | 70.8 | 65.3 | – | – | 75.3 | 71.1 |
| Mask3D | 56.6 | 68.4 | 68.7 | 66.3 | 64.5 | 75.5 | 72.8 | 74.5 |
| QueryFormer | 57.7 | 69.9 | 70.5 | 72.2 | 62.0 | 73.3 | 72.7 | 73.4 |
| MAFT | – | 69.1 | – | – | – | – | – | – |
| Spherical Mask | 60.5 | 72.3 | 71.3 | 76.3 | 64.0 | 72.3 | 78.1 | 77.7 |
| IKEA (Ours) | **61.1** | **73.0** | 72.0 | **77.0** | **65.8** | **76.9** | **79.4** | **78.9** |

Table 3: Comparison of 3DIS performance with state-of-the-art approaches on the Scan-Net200 (Rozenberszki et al., 2022) validation set, which contains 200 categories.

| Method | mAP | $mAP_{50}$ | $mAP_{25}$ |
|---|---|---|---|
| SPFormer (Sun et al., 2023) | 25.2 | 33.8 | 39.6 |
| Mask3D (Schult et al., 2022) | 27.4 | 37.0 | 42.3 |
| QueryFormer (Lu et al., 2023) | 28.1 | 37.1 | 43.4 |
| MAFT (Lai et al., 2023) | 29.2 | 38.2 | 43.3 |
| IKEA (Ours) | **29.9** | **38.9** | **44.9** |

Table 4: Comparison of 3DIS performance with state-of-the-art approaches on the STPLS3D (Chen et al., 2022) test dataset, including large-scale aerial outdoor scenes.

| Method | mAP | $mAP_{50}$ | $mAP_{25}$ |
|---|---|---|---|
| PointGroup (Jiang et al., 2020) | 23.3 | 38.5 | 48.6 |
| HAIS (Chen et al., 2021) | 35.1 | 46.7 | 52.8 |
| SoftGroup (Vu et al., 2022) | 47.3 | 63.1 | 71.4 |
| Mask3D (Schult et al., 2022) | 63.4 | 79.2 | 85.6 |
| IKEA (Ours) | **64.9** | **81.2** | **87.6** |

2022), S3DIS (Armeni et al., 2016), and STPLS3D (Chen et al., 2022). These four datasets provide 3D point cloud scan data collected in diverse real-world environments. Note that detailed descriptions of each dataset and all implementation details are provided in Appendix A.

**Evaluation Metrics.** We evaluate the 3D Instance Segmentation (3DIS) performance using the Average Precision (AP), a conventional metric in computer vision tasks. We report the mean average precision (mAP) across IoU (Intersection of Union) thresholds incremented by 5%, ranging from 50% to 95%. Also, we assess $mAP_{50}$ and $mAP_{25}$, representing accuracy with IoU thresholds of 50% and 25%, respectively. For the S3DIS (Armeni et al., 2016) dataset, we further provide mean precision (mPrec) and mean recall (mRec) with an IoU threshold of 50%, following previous methods.

## 4.2 Performance Comparison with State-of-the-Art Methods

In this section, we quantitatively compare our proposed framework IKEA with existing state-of-the-art methods. Despite significant improvements with high scores in 3DIS, these methods, regardless of the architecture, still waste considerable instance-wise candidates that may contain valuable information. To maximize the benefits of using numerous candidates, we present novel instance-wise knowledge aggregation (IKA) and guidance (ISG) networks. As shown in Tab. 1, we first evaluate mean average precision (mAP) with different IoU thresholds across 18 classes on the ScanNetV2 (Dai et al., 2017) validation set. IKEA generally outperforms other methods, achieving new state-of-the-art accuracy in terms of mAP, $mAP_{50}$, and $mAP_{25}$ (62.9 / 81.8 / 88.7). In Tab. 2, we assess 3DIS performance on Area 5 and the 6-fold cross-validation set of the S3DIS (Armeni et al., 2016) dataset. For Area 5 evaluation, we train with Areas 1 to 6, excluding 5, and validate with Area 5; in 6-fold cross-validation, we average the validation scores across all 6 areas. Our method demonstrates impressive performance in both evaluations, reaching 61.1 / 73.0 (Area 5) and 65.8 / 76.9 (6-fold) in mAP / $mAP_{50}$. Also, in Tab. 3, IKEA achieves robust scores on the ScanNet200 (Rozenberszki et al., 2022) dataset containing fine-grained 200 categories. Remarkably, as shown in Tab. 4, IKEA surpasses existing methods on the STPLS3D (Chen et al., 2022) dataset, including outdoor 3D scenes, with improvements of up to +1.5

Table 5: Ablation study to see the effect of two main modules based on kernel-based pipeline (K).

| Method (**Kernel-based**) | | | ScanNet Val | S3DIS Area 5 |
|---|---|---|---|---|
| Baseline (K) | IKA | ISG | mAP / mAP$_{50}$ | mAP / mAP$_{50}$ |
| ✓ | - | - | 56.8 / 73.3 | 56.3 / 67.5 |
| ✓ | ✓ | - | 62.7 / 80.8 | 60.4 / 72.6 |
| ✓ | - | ✓ | 62.8 / 81.1 | 60.6 / 71.9 |
| ✓ | ✓ | ✓ | **62.9 / 81.8** | **61.1 / 73.0** |

Table 6: Ablation study to see the effect of two main modules based on transformer pipeline (T).

| Method (**Transformer**) | | | ScanNet Val | S3DIS Area 5 |
|---|---|---|---|---|
| Baseline (T) | IKA | ISG | mAP / mAP$_{50}$ | mAP / mAP$_{50}$ |
| ✓ | - | - | 58.4 / 75.9 | 56.6 / 68.4 |
| ✓ | ✓ | - | 59.1 / 76.2 | 58.8 / 70.7 |
| ✓ | - | ✓ | 59.4 / 77.2 | 58.4 / 70.1 |
| ✓ | ✓ | ✓ | **60.8 / 77.9** | **59.0 / 71.6** |

/ 2.0 / 2.0 in mAP / mAP$_{50}$ / mAP$_{25}$. These results highlight that IKEA successfully handles instance candidates, improving their informativeness, as intended, in understanding real-world instances from both indoor and outdoor environments. In each table, "-" indicates unreported scores.

## 4.3 ABLATION STUDIES

**Effect of IKA and ISG Networks.** We evaluate variants of our method w/ and w/o IKA and ISG modules based on kernel (K) and transformer-based (T) architectures. As shown in Tab. 5 and 6, the addition of each module surpasses the score of baseline models (Ngo et al., 2023; Lai et al., 2023; Schult et al., 2022) across all experiments, regardless of model structure. Specifically, IKA boosts performance, with gains of up to +5.9 / 7.5 for ScanNetV2 (Dai et al., 2017) and +4.1 / 5.1 for S3DIS (Armeni et al., 2016) in mAP / mAP$_{50}$, underscoring the importance of integrating scattered clues from multiple candidates. Also, ISG contributes to the spatial understanding of instance candidates via intelligent structural guidance from clarity-enhanced features, improving up to +6.0 / 7.8 and +4.3 / 4.4. Eventually, utilizing all modules together gains further advancements, resulting in the best performance with notable progress on both structures. These results demonstrate each module plays a meaningful role in improving the discriminative ability of instance-wise features.

Furthermore, to assess how each module contributes to accurately determining the coverage of instance masks, we utilize the IoU metric, which is closely associated with segmentation performance. IoU measures the overlap between the predicted and ground truth masks by calculating the ratio of their intersection to their union. In Tab. 7, we compare the average IoU of our predicted instance masks and those of the baselines for both architectures. We excluded instance masks for walls and floors, as these are not considered in the overall score evaluation.

Table 7: Ablation study to see the effect of two main modules across both pipelines, using the IoU score on the ScanNetV2 (Dai et al., 2017) val set.

| Method | | | **Kernel-based** | **Transformer** |
|---|---|---|---|---|
| Baseline | IKA | ISG | Average IoU | Average IoU |
| ✓ | - | - | 0.8627 | 0.8486 |
| ✓ | ✓ | - | 0.8828 | 0.8728 |
| ✓ | - | ✓ | 0.8825 | 0.8730 |
| ✓ | ✓ | ✓ | **0.8904** | **0.8803** |

As shown in Fig. 1 (a) and Fig. 9, for previous works, a single instance is often separated into several fragments. These fragments potentially cause low IoU scores. To address this challenge, IKA encourages the network to understand the full coverage of a single instance by applying knowledge aggregation. The IoU improvements in using IKA across both architectures confirm that our approach effectively minimizes the negative impact of multiple fragments and enhances the accuracy of instance mask coverage. Also, the IoU tends to be low, particularly when instance masks include the surrounding backgrounds or when adjacent instances are not clearly distinguished, as shown in Fig. 1 (b) and Fig. 10, leading to overestimated or underestimated instance masks. To tackle these confusions, ISG guides the instance candidates with structural guidance from noise-reduced features. In addition to IKA, ISG enhances the IoU scores, verifying our guidance is sufficiently practical. We further quantitatively and qualitatively validate the significance of ours in Appendix C.2 and C.4.

**Correlation Regularization Terms.** Inspired by the self-supervised mechanisms (Zbontar et al., 2021), we softly guide highly correlated instance-wise features to be closer to each other in the latent space, enhancing their solidarity (Eq. 6 and Eq. 10). Specifically, we utilize dynamically generated pseudo-binary labels to regularize correlations. This element-wise regularizing strategy is conceptually comparable to

Table 8: Ablation study to compare our correlation regularization terms with usual cross-entropy loss.

| Method | ScanNet Val | S3DIS Area 5 |
|---|---|---|
| | mAP / mAP$_{50}$ | mAP / mAP$_{50}$ |
| Baseline (Ngo et al., 2023) | 56.8 / 73.3 | 56.3 / 67.5 |
| IKEA w/ Cross Entropy | 61.2 / 79.7 | 58.6 / 70.4 |
| IKEA | **62.9 / 81.8** | **61.1 / 73.0** |

Table 9: Ablation study to investigate the correlation matrix threshold $\tau$ for determining instance-wise candidates representing the same instance.

| Architecture | Threshold $\tau$ | ScanNet Val mAP / mAP$_{50}$ | S3DIS Area 5 mAP / mAP$_{50}$ |
|---|---|---|---|
| Baseline (K) | - | 56.8 / 73.3 | 56.3 / 67.5 |
| Kernel-based | 0.6 | 53.6 / 73.4 | 55.3 / 65.5 |
| Kernel-based | 0.7 | 57.2 / 76.4 | 59.1 / 69.2 |
| Kernel-based | 0.8 | 61.1 / 79.9 | 60.7 / 72.3 |
| Kernel-based | 0.9 | **62.9 / 81.8** | **61.1 / 73.0** |
| Baseline (T) | - | 58.4 / 75.9 | 56.6 / 68.4 |
| Transformer | 0.6 | 57.9 / 75.8 | 56.3 / 67.5 |
| Transformer | 0.7 | 59.2 / 76.2 | 57.4 / 68.6 |
| Transformer | 0.8 | 59.9 / 77.1 | 58.6 / 70.4 |
| Transformer | 0.9 | **60.8 / 77.9** | **59.0 / 71.6** |

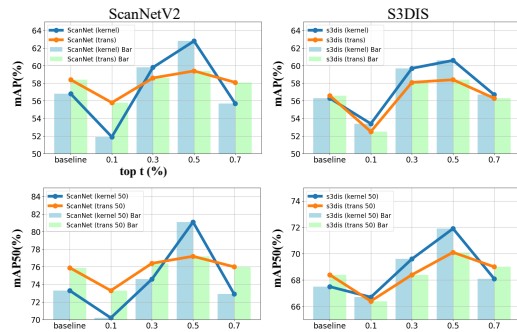

Figure 5: Performance comparison to explore the optimal top $t$ value for reducing ambiguity while keeping important cues of instance features.

standard cross-entropy loss, which measures the difference between predicted probabilities and ground-truth distributions. Therefore, to compare the two loss functions, we replace our regularization loss ($\mathcal{L}_{KA}$ and $\mathcal{L}_{SG}$) with standard cross-entropy loss. Here, we set all other settings constant. In Tab. 8, both approaches outperform the baseline (Ngo et al., 2023); however, IKEA with cross-entropy loss yields lower performance than the original IKEA. We consider that this performance gap comes from the differences in how each loss handles negative pairs. Cross-entropy loss aims to minimize the disparity between predictions and true labels without explicitly addressing negative pairs. On the other hand, ours considers both positive and negative pairs, reducing the position-wise distances between predicted and target matrices. This strategy encourages the model to establish valuable connections among highly correlated candidates, while minimizing confusion from irrelevant knowledge of unrelated candidates. In conclusion, these results validate the effectiveness of our regularization.

**Thresholds $\tau$ for Instance-wise Identification.** To enhance the informativeness of instance features, we utilize IKA and ISG networks. In the IKA, we compute the correlation matrix $I_{ka}$ between $F_{inst}$ and $F_{inst}^T$, while in the ISG, we calculate $I_{sg}$ between $F_{inst}$ and $\tilde{F}_{inst}$. Then, we specify candidates representing the same instance using predefined threshold $\tau$. In Tab. 9, based on kernel (K) and transformer (T) architectures (Ngo et al., 2023; Lai et al., 2023; Schult et al., 2022), we explore the impact of $\tau$ for useful knowledge interaction. When $\tau$ is lower than 0.7, the performance gain is insufficient for both architectures. Due to the inclusion of inaccurate information from different instance representative features, scores even decrease. But, with more precise identification using the higher $\tau$ ($\geqslant 0.8$), our knowledge interaction yields considerable improvements. This result emphasizes that the knowledge from accurately identified candidates is precious for enhancing instance interpretation. We also examine $\tau$ for the STPLS3D (Chen et al., 2022) dataset in Appendix C.3.

**Analysis of the Top $t$ values.** In this work, we utilize a truncated SVD algorithm (Hansen, 1987) to reduce inherent ambiguity from 3D features, clarifying the structural appearance of instances. We truncate the top $t$ vectors of $(U, \Sigma, V)$ and reconstruct $\tilde{F}'_{inst}$ using truncated $(\tilde{U}, \tilde{\Sigma}, \tilde{V})$. Here, the $t$ value impacts the balance between data compression and information preservation. Therefore, it is crucial to find the optimal value of $t$ for proper balance. To this end, we conduct experiments using various $t$ for both kernel (blue) and transformer (orange) architectures in Fig. 5. We represent $t$ as a percentage, so the x-axis indicates the proportion of vectors retained during truncation across three matrices. We observe that the trends of all experiments are similar. In settings with a lower $t$ percentage (0.1), the scores consistently decrease due to large information loss. However, with a higher $t$ (0.7), the effect of noise reduction is insufficient; thereby, performance is comparable to baselines (Ngo et al., 2023; Lai et al., 2023). We ultimately set $t$ as **0.5**, resulting in the highest accuracy while allowing us to keep meaningful information and effectively filter out inherent noises.

## 4.4 QUALITATIVE RESULTS

**Visual Comparison.** In Fig. 6, we qualitatively validate the effectiveness of our novel framework, IKEA. We visualize the predicted semantic (Sem.) and instance (Inst.) masks of the state-of-the-art kernel (K) (Ngo et al., 2023) and transformer (T) (Lai et al., 2023) baseline models with ours on the ScanNetV2 dataset. We also provide the corresponding ground truth for solid comparisons. As shown in Scene 1, ours accurately segments the *cabinet* as a single instance, whereas the baselines separate

Figure 6: Qualitative comparison of instance (Inst.) and semantic (Sem.) masks between the baselines (ISBNet (K) (Ngo et al., 2023), MAFT (T) (Lai et al., 2023)) and ours with Ground Truth masks on the ScanNetV2 val set. The critical differences are highlighted using yellow boxes. Note that the color map (right) represents semantic labels. We provide more qualitative results in the Appendix C.6.

it into multiple fragments. In Scene 2, ours correctly distinguishes multiple neighboring instances (four *chairs* and a *table*), while baselines erroneously recognize two *chairs* as one (K) or a *table* and a *chair* as one (T). Also, in Scene 3, where objects are disorderly adjacent, ours clearly captures the spatial range of the desk and chairs compared to the baselines. These qualitative results verify that our instance features contain highly informative cues for understanding complex 3D instances.

**Structural Guidance from $\tilde{F}_{inst}$.** In the ISG network, we guide the original instance features $F_{inst}$ to learn structural cues from the noise-reduced features $\tilde{F}_{inst}$. Here, in Fig. 7, we qualitatively confirm the significance of this knowledge. As shown in Scene 1 and 2, instance features (red) from $\tilde{F}_{inst}$ accurately capture the spatial range of instances (*chair* and *desk*), whereas instance features (blue) from $F_{inst}$ struggle to cover the complete coverage. Also, in Scene 3, where the distinction is unclear (blue) from the surroundings (*curtain* and *wall*), the candidate feature (red) from $\tilde{F}_{inst}$ clearly segments such instances. These findings highlight guidance from clarity-enhanced features is worthwhile.

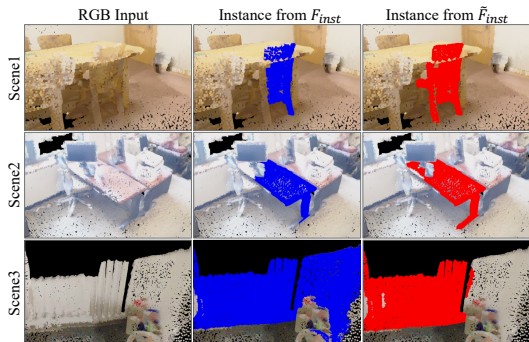

Figure 7: Instance feature visualizations of the original feature $F_{inst}$ and ambiguity-reduced $\tilde{F}_{inst}$.

## 5 DISCUSSION AND CONCLUSION

In this paper, we introduce IKEA, a novel instance-wise knowledge enhancement approach for the 3D Instance Segmentation task. We first focus on optimizing the efficiency of hundreds of instance candidates by effectively handling those representing the same single instance. To address these candidates, we propose an Instance-wise Knowledge Aggregation (IKA) to integrate spread clues using the correlation matrix. Moreover, we present an Instance-wise Structural Guidance (ISG), which enhances original instance features with fundamental cues for object understanding from ambiguity-reduced instance features using the truncated singular value decomposition. Our comprehensive experiments on large-scale benchmarks validate the effectiveness of our proposed methods, achieving new state-of-the-art results on both kernel and transformer architectures. Though IKEA significantly improves the 3D instance segmentation performance, it does not ensure perfect predictions for all 3D scenarios. Thus, a thorough plan is essential when implementing IKEA in contexts like autonomous driving or robotics navigation. In the future, we plan to further explore the potential of IKEA in various 3D perception tasks, such as 3D object detection (Qian et al., 2022; Chen et al., 2023) or 3D navigation (Liu et al., 2023; Zhang et al., 2023). It would be valuable to apply and validate our instance-wise knowledge enhancement approach with diverse models that utilize instance candidates. Further, we will investigate deep learning based denoising techniques (Tian et al., 2020; Elad et al., 2023) to reduce noises and highlight structural cues from instance-wise features.

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

APPENDIX

In this appendix, we provide further explanations and visualizations of our main paper, "Instance-wise Knowledge Enhancement for 3D Instance Segmentation". We first explain more details about the implementation and large-scale datasets (Appendix. A). Then, we describe the transformer-based IKEA architecture (Appendix. B). Also, we supply more quantitative and qualitative experimental results to validate the robustness of IKEA for 3D instance segmentation (Appendix. C).

## A EXPERIMENTAL SETUP

### A.1 DATASETS

We train and evaluate the overall performance using four landmark datasets for 3D instance segmentation: ScanNetV2 (Dai et al., 2017), ScanNet200 (Rozenberszki et al., 2022), S3DIS (Armeni et al., 2016), and STPLS3D (Chen et al., 2022).

**ScanNetV2.** The ScanNetV2 (Dai et al., 2017) dataset consists of high-quality, large-scale 3D point data with 1613 scenes from various room types, including bedrooms, libraries, and offices. It includes 1201 training scenes, 312 validation scenes, and 100 hidden test scenes. Each scene is captured with RGB-D cameras and categorized using 20 semantic classes and instance segmentation labels.

**ScanNet200.** To reflect diverse real-world scenarios, ScanNet200 (Rozenberszki et al., 2022) extends the original ScanNetV2 (Dai et al., 2017) dataset with fine-grained 200 categories. Scan-Net200 enables more practical assessments of how effectively methods can understand rare instances (*e.g.water cooler* or *keyboard piano*) and challenging, long-tail distribution scenes. In our experiments, we evaluate using 18 classes for ScanNetV2 and 198 classes for ScanNet200, excluding *wall* and *floor* categories.

**S3DIS.** The S3DIS (Armeni et al., 2016) dataset is large-scale benchmark, comprising a wide range of indoor environments, including 271 scenes from 6 areas within three different buildings. It is annotated with 13 semantic categories, and we utilize all these classes for evaluation. Following the standard validation protocol (Schult et al., 2022; Lai et al., 2023; Armeni et al., 2016), we report segmentation performance on Area 5 (the scenes in Area 5 for validation and the others for training) and 6-fold cross-validation (average across all 6 areas).

**STPLS3D.** The STPLS3D (Chen et al., 2022) dataset is a extensive aerial photogrammetry dataset containing both real and synthetic 3D point clouds. It includes 25 urban scenes covering 6 km², with 14 semantic classes. We use scenes 5, 10, 15, 20, and 25 for evaluation and the rest for training, following Vu et al. (2022); Chen et al. (2021).

### A.2 IMPLEMENTATION DETAILS

In this work, we implement our experimental setup using the PyTorch prevalent deep learning framework. For the kernel-based IKEA framework, we utilize two point aggregator blocks, each with a ball query radius of 0.2 and 0.4 and 32 neighbors for both layers. We also implement three dynamic convolution layers. We train our model for 120 epochs using a single RTX 3090 GPU 24GB ($\approx 16$ hours) with a batch size of 12 and applying the AdamW optimizer with a learning rate of $1 \times 10^{-3}$ and a weight decay of $1 \times 10^{-4}$. Furthermore, we set $\lambda$ parameters $[\lambda_{mask}, \lambda_{cls}, \lambda_{KA}, \lambda_{SG}]$ as $[5 \times 10^{-1}, 5 \times 10^{-1}, 1 \times 10^{-3}, 1 \times 10^{-3}]$. For the transformer-based IKEA pipeline, we utilize a transformer decoder with 6 layers and 8 heads to refine 400 instance queries. We use Fourier absolute position encoding with a temperature set to 10,000. For training, we train for 512 epochs with a batch size of 4, using the AdamW optimizer with a learning rate of $2 \times 10^{-4}$ and a weight decay of $5 \times 10^{-2}$ on a single RTX 3090 GPU 24GB ($\approx 24$ hours). We set $[\lambda_{mask}, \lambda_{cls}, \lambda_{KA}, \lambda_{SG}]$ as $[1, 1, 1 \times 10^{-3}, 1 \times 10^{-3}]$. Regardless of each architecture, we set the voxel size to 0.02m for the ScanNet (Dai et al., 2017) and S3DIS (Armeni et al., 2016) datasets, and 0.3m for the STPLS3D (Chen et al., 2022) dataset. During training, points are randomly sampled for augmentation with a maximum of 250,000 points, while all points are used for evaluation. This sampling technique is memory-efficient and can also serve as a dropout. Moreover, we set the correlation matrix threshold value $\tau$ to

Figure 8: An overview of transformer-based IKEA framework. Built upon the classic transformer-based structure, IKEA consists of four main modules: (1) Sparse Convolutional 3D Backbone; (2) Instance-wise Knowledge Aggregation (IKA); (3) Instance-wise Structural Guidance (ISG); and (4) Mask Transformer Decoder, which iteratively refines instance-wise queries to contain instance-specific information based on attention mechanisms and completes them into final instance masks.

0.9 (exceptionally 0.8 for STPLS3D) for precisely identifying instance-wise candidates representing the same instance, and the top $t$ value as 0.5 for an optimal balance between data compression and information preservation. Also, for $\mathcal{L}_{KA}$ and $\mathcal{L}_{SG}$, we set the balance hyperparameters $w$ and $\tilde{w}$ to $5.1 \times 10^{-3}$ following (Zbontar et al., 2021; Jang et al., 2024).

## B  TRANSFORMER-BASED IKEA FRAMEWORK

In this section, we describe our instance-wise knowledge enhancement methods based on classic transformer-based architecture (Schult et al., 2022; Lai et al., 2023). As illustrated in Fig. 8, our model consists of four main modules: (1) Sparse Convolutional 3D Backbone; (2) Instance-wise Knowledge Aggregation (IKA), which associates the scattered cues of the same single instance; (3) Instance-wise Structural Guidance (ISG), which enhances the spatial understanding of instance features using noise-reduced features; and (4) Mask Transformer Decoder, which refines hundreds of instance candidate queries to contain instance-specific information based on attention mechanisms.

**Transformer-based 3DIS Architecture.**  As in the kernel-based architecture (see Sec. 3), the sparse convolutional U-Net backbone (Graham et al., 2018) first takes a colored point cloud $P \in \mathbb{R}^{N_p \times 6}$ as input and extracts full-resolution feature maps $F'_p \in \mathbb{R}^{N_p \times D}$. We then produce $F'_p$ into mask features $F_{mask} \in \mathbb{R}^{N_p \times D}$ and point features $F_p \in \mathbb{R}^{N_p \times D}$ via MLP layers. Following Schult et al. (2022); Misra et al. (2021), we set zero-initialized non-parametric instance queries $Q \in \mathbb{R}^{N_k \times D}$, referring to point positions sampled with *furthest point sampling* (FPS) (Eldar et al., 1997). Given the $F_{mask}$, $F_p$ and $Q$, the transformer decoder layer iteratively enhances the queries $Q$ using attention mechanisms. Specifically, we employ the masked cross-attention with an intermediate foreground mask $\mathcal{M}_{attn}$. We compute the similarity between $Q$ and $F_{mask}$ using the dot product operation, then calculate the probability of the instance mask using the sigmoid function as follows:

$$\mathcal{M}_{attn} = \{m_{i,j} = [\sigma(F_{mask} \cdot Q^T)_{i,j} > 0.5]\} \tag{12}$$

where the threshold value is 0.5 for binary attention mask. With $\mathcal{M}_{attn}$, $Q$ attends to point features $F_p$ in the cross-attention layer to contain instance-specific information as follows:

$$Q = \text{softmax}(\mathbf{Q}\mathbf{K}^T/\sqrt{D} + \mathcal{M}_{attn})\mathbf{V} \tag{13}$$

where $\mathbf{K}$ and $\mathbf{V}$ are linearly projected from $F_p$, and $\mathbf{Q}$ are from $Q$. Subsequently, we utilize the standard self-attention layer. Here, the queries, keys, and values are all linear projections of $Q$. After passing through these layers, we predict the final instance masks using the queries from the last layer.

**IKEA Approach.**  In transformer-based architecture, iterative decoder layers attend point features, which often contain inherent fuzzy noises due to the sparse and incomplete nature of point clouds.

Thus, repetitive layers can lead to noise accumulation in the candidate features during the attention operations, potentially resulting in spatial range misinterpretations. To tackle this challenge, we introduce our **ISG** network, which leverages a simple yet effective truncated SVD (Hansen, 1987) technique within the decoder layers to regularize the correlations between the original and clarity-enhanced query features. Then, the iterative layers continuously enrich instance candidate queries with structural cues, as detailed in Sec. 3.3. Also, we implement our **IKA** network, which is designed to integrate scattered clues across query features representing the same single instance. The IKA optimizes correlations among instance candidate queries, as outlined in Sec. 3.2. Ultimately, IKEA predicts more accurate instance segmentation masks with highly informative instance query features.

## C EXPERIMENTAL RESULTS

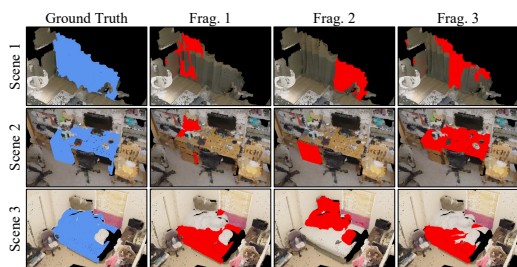
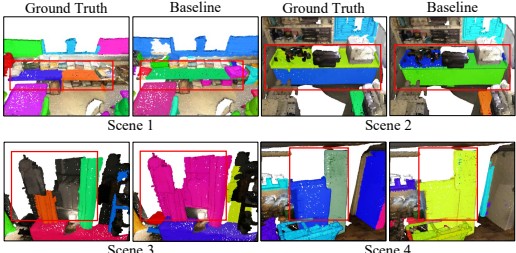

Figure 9: Additional challenging cases from prior works. Instance-wise candidates usually represent incomplete fragments of the same single instance.

Figure 10: Additional challenging cases from prior works. They often confuse instances from backgrounds or misunderstand the spatial range.

### C.1 PRIOR WORKS IN TABLE 1 AND 2

In Tab. 1 and Tab. 2, we quantitatively compare our proposed framework IKEA with various existing state-of-the-art methods. IKEA generally outperforms other approaches, including GSPN (Yi et al., 2019), SGPN (Wang et al., 2018), ASIS (Wang et al., 2019), 3D-Bonet (Yang et al., 2019), 3D-SIS (Hou et al., 2019), MTML (Lahoud et al., 2019), 3D-MPA (Engelmann et al., 2020), DyCo3D (He et al., 2021), PointGroup (Jiang et al., 2020), MaskGroup (Zhong et al., 2022), OccuSeg (Han et al., 2020), SSTNet (Liang et al., 2021), SoftGroup (Vu et al., 2022), Mask3D (Schult et al., 2022), DKNet (Wu et al., 2022), QueryFormer (Lu et al., 2023), ISBNet (Ngo et al., 2023), MAFT (Lai et al., 2023), and Spherical Mask (Shin et al., 2023), demonstrating impressive advancements.

### C.2 EFFECTIVENESS OF THE IKA

To further validate the effectiveness of our IKA network, we investigate the average variance and standard deviation of instance candidate features across both kernel and transformer architectures of baselines (Ngo et al., 2023; Lai et al., 2023) and those with IKA in Tab. 10 and Fig. 11. We first identify candidates representing the same instance using ground-truth instance masks to ensure fair and more precise comparisons between predicted instance masks from each model. We then calculate the variance and standard deviation of features corresponding to identical instances. Compared to baselines, incorporating

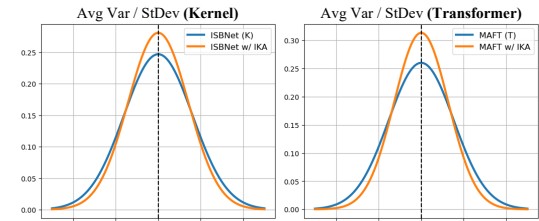

Figure 11: Distribution of the average variance and standard deviation of instance features across baselines (blue) and those with IKA (orange) network.

IKA consistently achieves lower variance and standard deviation, regardless of the architecture. These results verify that our instance-wise aggregation approach effectively enhances the correlations between candidates from the same instance, establishing meaningful associations.

Table 10: Average variance and standard deviation of instance features across kernel and transformer-based models and those with IKA.

| Method | Avg Variance | Avg StDev |
|---|---|---|
| ISBNet (K) (Ngo et al., 2023) | 2.8441 | 1.6152 |
| ISBNet w/ IKA | **2.4620** | **1.4199** |
| MAFT (T) (Lai et al., 2023) | 2.3613 | 1.5352 |
| MAFT w/ IKA | **2.1278** | **1.2743** |

Table 11: Analysis of the correlation matrix threshold $\tau$ for instance-wise candidate identification on the STPLS3D (Chen et al., 2022) dataset.

| Architecture | Threshold $\tau$ | mAP | mAP$_{50}$ |
|---|---|---|---|
| Baseline Schult et al. (2022) | - | 63.4 | 79.2 |
| IKEA | 0.6 | 61.8 | 78.8 |
| IKEA | 0.7 | 64.5 | 80.8 |
| IKEA | 0.8 | **64.9** | **81.2** |
| IKEA | 0.9 | 64.7 | 81.0 |

## C.3 Threshold $\tau$ for the STPLS3D Dataset

In addition to analyzing threshold $\tau$ on the ScanNetV2 (Dai et al., 2017) and S3DIS (Armeni et al., 2016) datasets presented in Tab. 9, we also conduct experiments across a range of $\tau$ values to find the optimal threshold that facilitates robust identification of candidates representing the same single instance for the STPLS3D (Chen et al., 2022) dataset. As shown in Tab. 11, the model effectively determines instance candidates likely to represent the same instance with a somewhat lower threshold of 0.8 for the STPLS3D, compared to 0.9 for both ScanNetV2 and S3DIS. This difference is probably because STPLS3D includes relatively monotonous large instances, such as buildings and cars, unlike ScanNetV2 or S3DIS, which contain more complex indoor props.

## C.4 Effectiveness of the IKEA framework

We provide t-SNE (Van der Maaten & Hinton, 2008) visualizations of instance candidate features clustered using the density-based spatial clustering (DBSCAN) (Ester et al., 1996) algorithm to further qualitatively demonstrate the significance of the IKEA framework. As shown in Fig. 12, the candidate features in the baseline method (Ngo et al., 2023) are wildly scattered without patterns in the feature space, resulting in multiple fragments. In contrast, IKEA produces relatively distinctive clusters for the same scene, with clusters that are accurate to the number of instances. These qualitative findings confirm that our IKA and ISG networks handle hundreds of instance candidate features effectively.

## C.5 Discussion on the number of instances

We investigate the number of instances within scenes from various benchmarks, including ScanNetV2 (Dai et al., 2017), S3DIS (Armeni et al., 2016), and STPlS3D (Chen et al., 2022). We randomly sampled around 30% of scenes from each dataset and computed the minimum, maximum, and average number of instances. On average, the S3DIS dataset, which consists of a

Table 12: Minimum, maximum, and average number of instances per scene from various datasets.

| Dataset | min Inst. | max Inst. | avg Inst. |
|---|---|---|---|
| ScanNetV2 (Dai et al., 2017) | 3 | 104 | 15.2 |
| S3DIS (Armeni et al., 2016) | 6 | 90 | 34.5 |
| STPLS3D (Chen et al., 2022) | 2 | 93 | 25.2 |

wide range of indoor environments such as exhibition and educational spaces, includes more instances (34.5) per scene than the other two datasets. The ScanNetV2, containing rooms of various sizes, from small bathrooms to large conference rooms, has relatively fewer average instances (15.2) per scene but occasionally includes the maximum number (104) of instances. In Fig. 13, we also visualize global view examples of scenes, especially with a large number of instances from each dataset. Since our methods regularize features based on correlations among all hundreds of instance candidates, IKEA is effective regardless of instance numbers. Although it might be less effective in extreme cases where the number of objects in a scene exceeds hundreds of candidates, these scenarios are rare.

## C.6 Visual Comparison

In this section, we present additional qualitative visualization results of our framework, IKEA, compared to existing state-of-the-art models: ISBNet (kernel-based, K) (Ngo et al., 2023) and MAFT (transformer-based, T) (Lai et al., 2023), in Fig. 14 and Fig. 15. We visualize the predicted semantic (Sem.) and instance (Inst.) results with corresponding ground truth on the ScanNetV2 (Dai et al., 2017) validation set, using red colored boxes to highlight the critical differences for better comparison. First, as shown in Fig. 14, our method outperforms existing methods in precisely

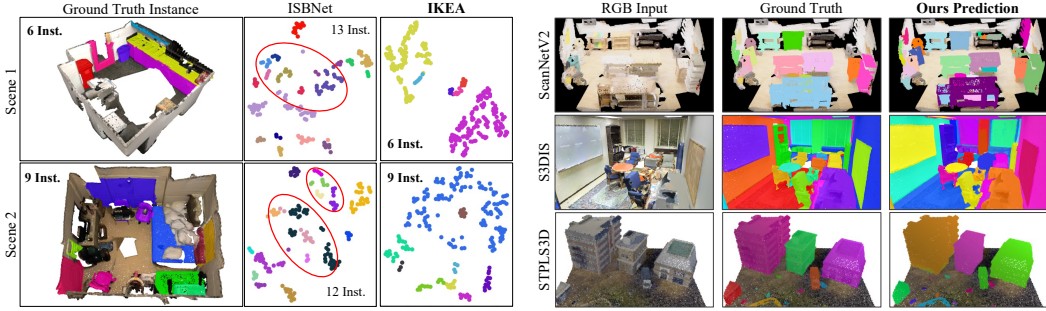

Figure 12: t-SNE (Van der Maaten & Hinton, 2008) visualization of instance candidate features from kernel-based baseline (ISBNet) and IKEA.

Figure 13: Global view visualizations of instance masks with ground truth from scenes containing a large number of instances across various datasets.

classifying a single instance into one category without fragments. In particular, compared to baseline models, IKEA more accurately identifies large instances like a *sofa* (Scene 4) or *cabinet* (Scene 5). Furthermore, as shown in Fig. 15, ours consistently distinguishes single objects as a whole unit, unlike the baselines, which often erroneously segment multiple fragments. For example, in Scenes 12-14, where objects are closely adjacent, ours clearly captures their spatial range. These outcomes underscore the effectiveness of our proposed modules, which enhance instance-wise knowledge to comprehend complex real-world situations.

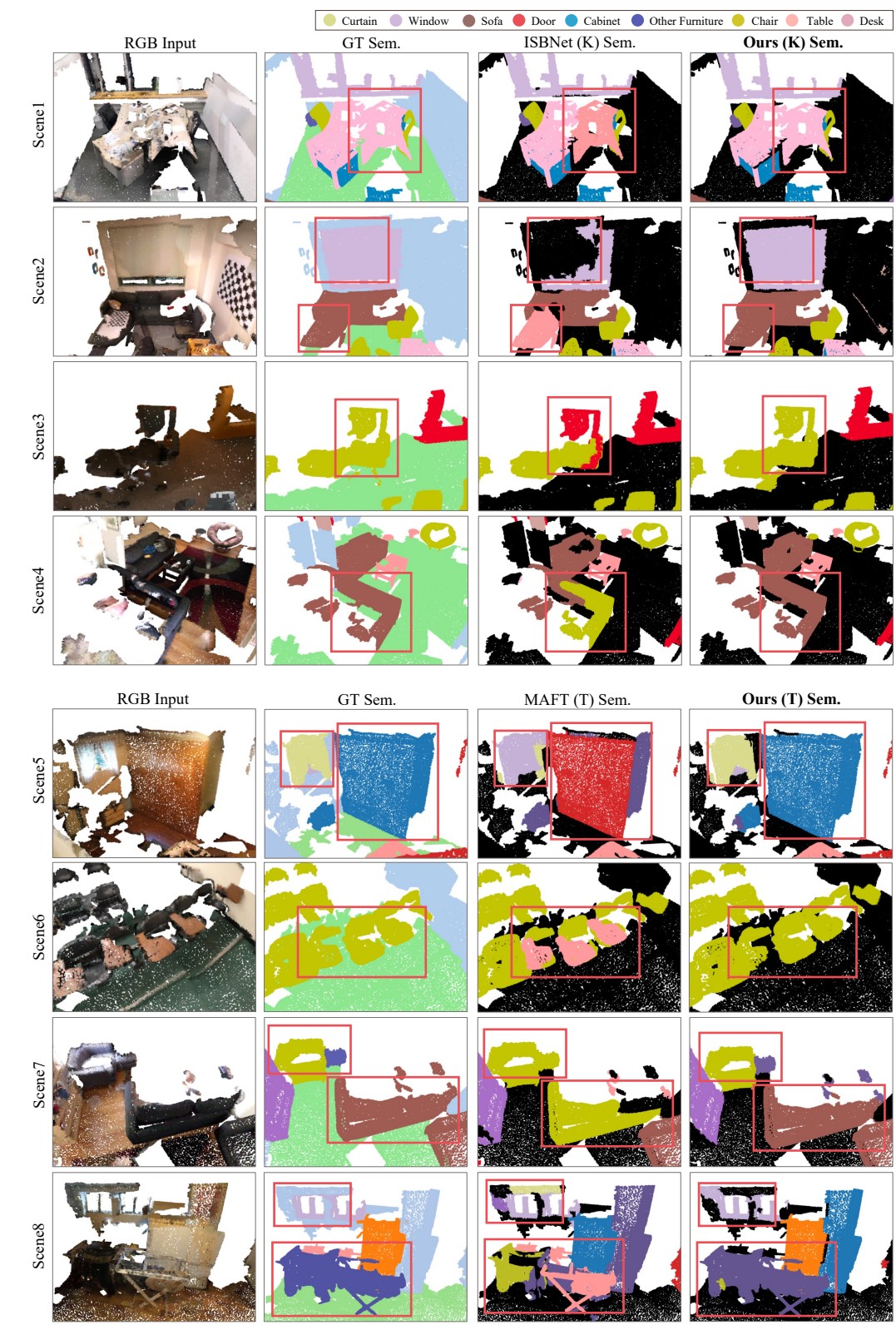

Figure 14: Qualitative comparisons of 3D Instance Segmentation performance on the ScanNetV2 (Dai et al., 2017) validation set. We visualize semantic (Sem.) masks of ISBNet (kernel-based, K) (Ngo et al., 2023), MAFT (transformer-based, T) (Lai et al., 2023) and ours based on both architecture with Ground Truth (GT) masks. The critical differences are highlighted using red-colored boxes for better comparison. Note that the color map (top right) represents semantic labels.

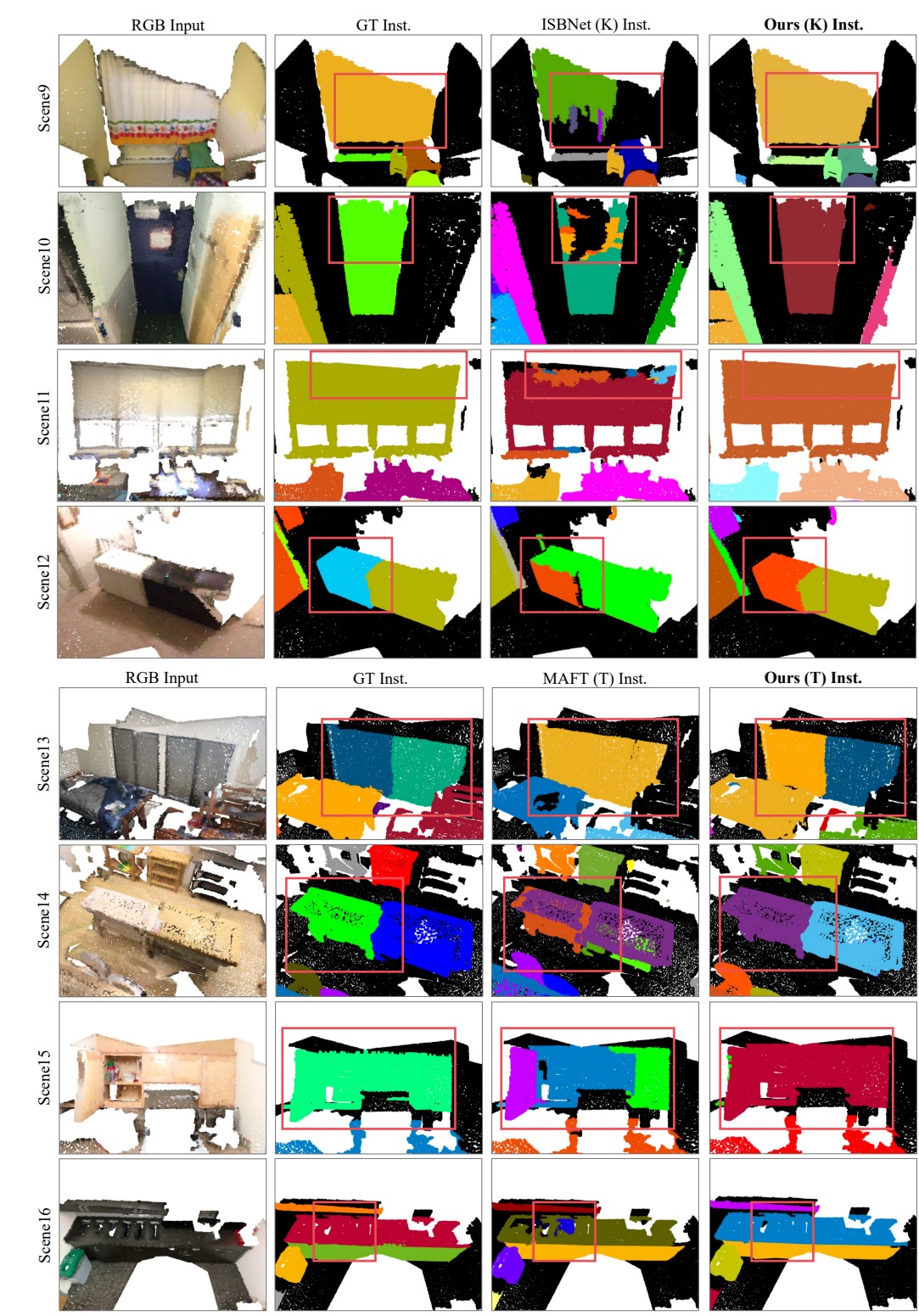

Figure 15: Qualitative comparisons of 3D Instance Segmentation performance on the ScanNetV2 (Dai et al., 2017) validation set. We visualize instance (Inst.) masks of ISBNet (kernel-based, K) (Ngo et al., 2023), MAFT (transformer-based, T) (Lai et al., 2023) and ours based on both architecture with Ground Truth (GT) masks. The critical differences are highlighted using red-colored boxes for better comparison.

