# OpenReview forum: "Instance-wise Knowledge Enhancement for 3D Instance Segmentation"
_ICLR.cc/2025/Conference — ICLR 2025 Conference Withdrawn Submission_

### Official Review · Reviewer_zoMM · 2024-10-24

**Soundness:** 2
**Presentation:** 3
**Contribution:** 2
**Rating:** 5
**Confidence:** 4

**Summary:**

The paper focuses on 3D instance segmentation, particularly addressing the prevalent paradigm of instance-wise candidates. It identifies two key problems overlooked by previous methods. Firstly, valuable cues encoded in multiple candidates representing different parts of the same instance often result in fragmented instance masks. Secondly, there is a failure to capture the precise spatial extent of complex 3D instances due to inherent fuzzy noise from sparse and unordered point clouds. To tackle these challenges, the paper proposes two modules: Instance-wise Knowledge Aggregation and Instance-wise Structural Guidance. Experimental results on ScanNetV2, ScanNet200, S3DIS and STPLS3D demonstrate that the proposed approach outperforms existing methods.

**Strengths:**

1. The writing is clear and easy to understand.
2. The proposed method demonstrates better performance across several relevant datasets.

**Weaknesses:**

1. **Instance-wise Knowledge Aggregation**: According to equations (5) and (6), and Table 9, the performance is best when the threshold τ is set to 0.9. This means that the element-wise correlation regularization loss $L_{ka}$ will constrain $I_{ka}(i, j)$ to 1 for values greater than 0.9, and 0 for the rest. You need to more clearly show the relationships between instances with correlations greater than 0.9 and those with correlations less than 0.9. A statistical analysis is needed. Additionally, constraining $I_{ka}(i, j)$ to 0 for values less than 0.9 might have negative impacts. Since 0.9 is a high threshold, the instances with values greater than 0.9 are already very similar, so the positive gains might be limited. Considering that all values less than 0.9 are constrained to 0, could the negative impacts outweigh the positive gains?

2. **Instance-wise Structural Guidance**:

   **a**. Regarding the definition of noise: In lines 292-293, it is stated that “the higher singular values in Σ include meaningful information for object cognition, while the lower-rank vectors are regarded as less important, potentially causing ambiguity.” This point is hard to convince me. Lower-rank vectors might also contain unique and distinguishing features of an instance and might not necessarily be less important. If you want to prove this, you need to provide experimental evidence.

   **b**. Regarding equations (9) and (10): These equations are very similar to equations (5) and (6). From the experimental results in Table 5, it can also be observed that IKA and ISG individually bring significant improvements (rows 1, 2, and 3). However, when combined (rows 2, 3, and 4), the improvements are relatively limited. This suggests that these two modules might have similar functions. I can understand that the diagonal elements of the matrix can reduce noise, but I don’t understand how the off-diagonal elements can achieve noise reduction. Moreover, in this case as well, τ=0.9 performs best. Could the negative impacts outweigh the positive gains?

4. A third major issue I see with this paper is the extensive number of detailed "ablation studies," which I would rather call hyperparameter tuning on the validation set. All results are shown on public validation sets where the hyperparameters have also been optimized. From Table 9, I find that the threshold τ has a significant impact on the results. For example, when τ is 0.6, the performance is lower than the baseline. This raises the question of whether the chosen value of τ is robust enough for new datasets. I have similar concerns regarding the value of $t$ used in Figure 5.

5. Results on ScannetV2 test set and Scannet++ test set are very meaningful.

**Questions:**

See weaknesses above.

---

### Official Review · Reviewer_c9Dz · 2024-10-24

**Soundness:** 3
**Presentation:** 3
**Contribution:** 3
**Rating:** 6
**Confidence:** 3

**Summary:**

The manuscript introduces IKEA (Instance-Wise Knowledge Enhancement Approach), a framework designed to enhance the performance of 3D instance segmentation by improving the informativeness of instance-wise candidates.

This work claims that most recent 3D instance segmentation methods generate hundreds of instance candidates, but they fail to fully utilize the rich instance-specific information, resulting in fragmented masks or imprecise segmentations. IKEA addresses this by introducing two key modules: Instance-Wise Knowledge Aggregation (IKA) and Instance-Wise Structural Guidance (ISG).

The IKA module optimizes correlations among multiple candidates representing the same instance, allowing for the aggregation of scattered knowledge from different parts of an object. By identifying highly correlated instance features, IKA integrates them into a more complete and cohesive representation of each instance. This helps reduce fragmented masks and enhances the model’s understanding of complex 3D structures.

The ISG module improves the structural understanding of instances by using a truncated Singular Value Decomposition (SVD) technique. ISG reduces inherent noise in 3D features, helping the model focus on the essential spatial cues of objects. This structural guidance is especially useful in complex scenes where objects may overlap or be partially occluded.

The authors demonstrate IKEA’s effectiveness through extensive experiments on four large-scale 3D instance segmentation benchmarks: ScanNetV2, ScanNet200, S3DIS, and STPLS3D. The results show that IKEA outperforms state-of-the-art methods across all these datasets, achieving significant improvements in metrics like mAP, mAP50, and mAP25. The method is also validated on both kernel-based and transformer-based architectures, showcasing its versatility and scalability.

**Strengths:**

(+) The idea of improving the informativeness of instance candidates by aggregating scattered knowledge and guiding candidates with structural cues is new and addresses common weaknesses in current 3D instance segmentation methods.

(+) The use of a truncated SVD algorithm for noise reduction in 3D features is a good contribution, as it allows for better structural understanding of objects, leading to more precise instance segmentation.

(+) The experimental results on large-scale benchmarks show that IKEA achieves state-of-the-art performance, especially on complex datasets like ScanNet200 and STPLS3D.

**Weaknesses:**

(-) Although the proposed method achieves superior performance, there is limited discussion on its computational complexity. Given that IKEA integrates two additional modules (IKA and ISG), it would be important to analyze the trade-off between performance improvement and computational cost, especially for real-time applications like robotics or autonomous driving.

(-) The manuscript lacks a thorough ablation study comparing the performance impact of IKA and ISG separately on different types of scenes (e.g., highly cluttered versus simple environments). This could provide a better understanding of when each module is most beneficial.

(-) The use of the name “IKEA” for the proposed method may raise concerns since IKEA is a well-known trademark for a multinational furniture company. The authors are encouraged to consider a different, more appropriate name that does not conflict with established trademarks.

---

### Justification of rating
While this manuscript presents a novel approach with the IKEA framework, combining Instance-Wise Knowledge Aggregation (IKA) and Instance-Wise Structural Guidance (ISG) to improve 3D instance segmentation, there are certain aspects that could be refined for a stronger impact.

The proposed modules address common issues in 3D instance segmentation, such as fragmented masks and lack of structural coherence, which is an important advancement. Additionally, the experimental results on multiple large-scale benchmarks provide convincing evidence of the method’s effectiveness.

However, I have some concerns regarding the computational complexity introduced by the two additional modules (IKA and ISG). A more in-depth discussion of computational overhead and the trade-offs between performance gains and computational costs would help clarify its practicality, especially in real-time applications. Furthermore, a more thorough ablation study to isolate the contributions of IKA and ISG would provide a better understanding of their individual effectiveness.

The use of the name “IKEA” for the method could be problematic due to its association with a well-known trademark, and the authors should consider renaming it to avoid legal issues.

Lastly, while I believe the paper brings valuable contributions to the field, I do not directly work on this topic and will leave more room for other reviewers to assess the technical novelty and depth of the contributions.

Overall, I would suggest a borderline accept rating based on the strengths and weaknesses highlighted above.

**Questions:**

- **Q1:** Could the authors provide more details regarding the computational overhead introduced by the IKA and ISG modules? Specifically, how does the addition of these two components affect the runtime performance of the model, and is it suitable for real-time applications like autonomous driving or robotics?

- **Q2:** While the manuscript shows strong performance improvements, could the authors conduct further ablation studies to analyze the individual contributions of IKA and ISG? Understanding how each module impacts segmentation performance in various scenes (e.g., highly cluttered vs. simple environments) would add valuable insights into their specific roles.

- **Q3:** The name “IKEA” may raise potential trademark issues, as it is already associated with a global furniture company. Would the authors consider selecting a different name to avoid confusion or potential legal complications?

- **Q4:** Could the authors discuss more about the scalability of IKEA on even larger and more diverse datasets? Specifically, how does the model perform on outdoor, long-range, or dynamic scenes, which present different challenges compared to the indoor datasets used in the experiments?

---

### Official Review · Reviewer_TPrj · 2024-10-26

**Soundness:** 2
**Presentation:** 3
**Contribution:** 2
**Rating:** 5
**Confidence:** 4

**Summary:**

This paper proposes an instance-level knowledge enhancement approach for 3D instance segmentation. The key contributions of the paper include:

1. Instance-level Knowledge Aggregation: It optimizes the correlation between instance candidate features, linking dispersed information from multiple instance candidates to predict a complete mask for each instance.
2. Instance-level Structure Guidance: It leverages truncated Singular Value Decomposition (SVD) to emphasize essential cues within instance features while reducing noisy ambiguity.

**Strengths:**

1. Experimental results across multiple datasets demonstrate that IKEA achieves state-of-the-art performance in 3D instance segmentation tasks.
2. The paper is logically structured and easy to understand.

**Weaknesses:**

The two proposed modules, IKA and ISG, are not particularly novel, with the main difference being that ISG introduces SVD for feature denoising. As a result, the technical contribution of this work feels somewhat limited.

**Questions:**

1. Would applying a many-to-one matching between candidate points and ground truth allow the same ground truth constraint to be used for different candidates of the same instance? Could this achieve a similar feature-guidance effect as IKA during training?

2. Since IKA and ISG only consider feature similarity, could this lead to incorrect guidance of neighboring candidates that do not belong to the same instance?

---

### Official Review · Reviewer_SSno · 2024-10-27

**Soundness:** 2
**Presentation:** 3
**Contribution:** 2
**Rating:** 5
**Confidence:** 5

**Summary:**

This manuscript aims to alleviate the instance fragments and fuzzy noise issue in 3D point cloud instance segmentation. It proposes Instance-wise Knowledge Aggregation (IKA) to connect scattered instance-specific information of the same single instance by optimizing their correlations. Furthermore, it develops Instance-wise Structural Guidance (ISG), which leverages the SVD to emphasize essential cues from features while filtering out noises. Moreover, the proposed method achieve the state-of-the-arts performance on four public datasets (validation sets).

**Strengths:**

1. This manuscript propose Instance-wise Knowledge Aggregation (IKA) module to aggregate scattered instances
2. This manuscript propose Instance-wise Structural Guidance (ISG) module which utilizes SVD to filter out noise.
3. The proposed method achieve the state-of-the-arts performance on four public datasets (validation sets).

**Weaknesses:**

1.To ensure a fair comparison, the results of the proposed method on the hidden test sets of ScanNetv2 and ScanNet200 need to be provided, because validation set results could be overfitted with hyperparameters λ,τ, and t.

2.The reference list needs to be expanded and updated.

  1). The following works should be added in the related work and Tab.1&2: PBNet:Divide and Conquer: 3D Point Cloud Instance Segmentation With Point-Wise Binarization(ICCV23), OneFormer3D: One Transformer for Unified Point Cloud Segmentation(CVPR24), ODIN: A Single Model for 2D and 3D Segmentation(CVPR24),

 2). The following reference should be updated: Mask3D(ICRA23), Spherical Mask(CVPR24)

3.An analysis and comparison of inference time efficiency are missing.

4.There is a lack of analysis of how current SOTA works handle fragment instance challenges, for example, HAIS, DKNet, and PBNet all deal with fragments. What are the advantages of the proposed method?

**Questions:**

I recognize the contributions made by this manuscript, but I still have the following concerns:

1.Does the author have more specific data to support the optimization of their method for fragment instances, rather than just general mAP metrics? For example, Figure 4 in HAIS.

2.Does the semantic information of instances affect the Feature Embedding Space operation? For example, what would the t-SNE plot look like for two different chair instances within the same scene?

3.The data augmentation techniques used need to be specified in the implement details. Is mixup, ensemble, superpoint pooling, and other techniques employed?

4.The weights for Lka and Lsg losses are very small at 1e-3; I suspect that there is a lack of averaging after summing each instance losses.

---

### Official Review · Reviewer_XVw3 · 2024-11-04

**Soundness:** 2
**Presentation:** 3
**Contribution:** 2
**Rating:** 5
**Confidence:** 2

**Summary:**

This paper introduces IKEA, a novel approach for improving 3D instance segmentation by enhancing instance-wise knowledge. IKEA addresses two major challenges in existing 3DIS methods: fragmented instance masks and imprecise spatial range understanding due to noisy, unordered 3D point clouds. The proposed solution comprises two main components: Instance-wise Knowledge Aggregation (IKA) and Instance-wise Structural Guidance (ISG). IKA refines instance mask predictions by associating scattered instance-specific features, while ISG utilizes truncated Singular Value Decomposition (SVD) to reduce noise and provide structural cues.

**Strengths:**

- The figures provided in the paper are clear, effectively illustrating the idea of IKEA’s approach.
- The concept of using correlations among candidate instances to improve mask coherence is intelligently designed to address mask fragmentation issues.

**Weaknesses:**

- The algorithm’s complexity may make it challenging to understand and apply in practice.
- The use of self-supervised learning to obtain  $I_{ka}$ feels less convincing. In my opinion, it could be more accurately described as a form of regularization rather than self-supervised learning.

**Questions:**

While I am not deeply familiar with the 3D Instance Segmentation field, some questions arose:
- Given that ground-truth (GT) masks are available for each $F_{inst}$ and could be used to derive $I_{ka}$, why was a self-supervised approach chosen to obtain $I_{ka}$ instead?
- How does IKEA handle overlapping masks from multiple kernels $K$ in the output? Is Non-Maximum Suppression (NMS) applied, or is there a Hungarian matching loss similar to DETR during training?

---

### Note · Authors · 2024-11-14

I have read and agree with the venue's withdrawal policy on behalf of myself and my co-authors.